# *Ante-* and *post-mortem* cellular injury dynamics in hybrid poplar foliage as a function of phytotoxic O$_3$ dose

**Benjamin Turc** [1,2]*, **Yves Jolivet**[1], **Mireille Cabané**[1], **Marcus Schaub**[2], **Pierre Vollenweider**[2]

**1** UMR Silva, AgroParisTech, INRAE, University of Lorraine, Lorraine, France, **2** Swiss Federal Institute for Forest, Snow and Landscape Research WSL, Birmensdorf, Switzerland

* bturc2@unl.edu

**Data Availability Statement:** All relevant data are within the paper and its Supporting information files.

## Abstract

After reaching phytotoxic levels during the last century, tropospheric ozone (O$_3$) pollution is likely to remain a major concern in the coming decades. Despite similar injury processes, there is astounding interspecific–and sometimes intraspecific–foliar symptom variability, which may be related to spatial and temporal variation in injury dynamics. After characterizing the dynamics of physiological responses and O$_3$ injury in the foliage of hybrid poplar in an earlier study, here we investigated the dynamics of changes in the cell structure occurring in the mesophyll as a function of O$_3$ treatment, time, phytotoxic O$_3$ dose (POD$_0$), leaf developmental stage, and mesophyll layer. While the number of Hypersensitive Response-like (HR-like) lesions increased with higher O$_3$ concentrations and POD$_0$, especially in older leaves, most structural HR-like markers developed after cell death, independent of the experimental factors. The pace of degenerative Accelerated Cell Senescence (ACS) responses depended closely on the O$_3$ concentration and POD$_0$, in interaction with leaf age. Changes in total chlorophyll content, plastoglobuli and chloroplast shape pointed to thylakoid membranes in chloroplasts as being especially sensitive to O$_3$ stress. Hence, our study demonstrates that early HR-like markers can provide reasonably specific, sensitive and reliable quantitative structural estimates of O$_3$ stress for *e.g.* risk assessment studies, especially if they are associated with degenerative and thylakoid-related injury in chloroplasts from mesophyll.

## Introduction

Tropospheric ozone (O$_3$) concentrations have increased during the past century, as a consequence of increasing NO$_x$ and volatile organic compound (VOC) levels [1, 2]. Over the last decade, there has been a slight decrease in tropospheric O$_3$ levels in North America and Europe as a result of emission control [3, 4]. In contrast, ground-level concentrations of O$_3$ are still increasing in highly polluted Asian areas, due to progressing industrialization and urbanization [5–7]. As a consequence, crop yields decreased by 10–20%, depending on the species

**Funding:** This work was supported by the French National Research Agency (ANR, "Investissement d'avenir" from the program Lab of Excellence ARBRE: ANR-11-LABX-0002-01, to YJ), and by a Swiss Federal Institute for Forest, Snow and Landscape Research (WSL) internal grant (201701N1428, to PV). The funders had no role in study design, data collection and analysis, decision to publish, or preparation of the manuscript.

**Competing interests:** The authors have declared that no competing interests exist.

[8–10], and biomass production of trees decreased by 15% [11–13], leading to significant economic loss [14]. $O_3$ uptake and its impact on plants are mediated by leaf conductance and the opening of stomata [15, 16]. Accordingly, critical $O_3$ levels (CLs) for preventing adverse effects such as yield or biomass loss are based on $O_3$ flux assessments and have been defined for several plant types and groups [17]. However, the relationship between such metrics and effective $O_3$ impacts on physiological and cellular processes has remained poorly defined and understood so far [18].

As hypothesized by Vollenweider, Günthardt-Goerg [19] on the basis of converging evidence [20–24], Turc, Vollenweider [18] recently demonstrated that the distinct responses to $O_3$ stress within leaf tissues develop according to specific dynamics. They further highlighted the importance of environmental conditions regarding $O_3$ responses, given the close dependency between response order and illumination conditions. Reaching the leaf apoplast, $O_3$ quickly breaks down, causing the formation of reactive oxygen species (ROS) and cascades of oxidative bursts [25, 26]. If a threshold within the mesophyll symplast is exceeded, oxidative stress triggers the formation of necrotic lesions, resulting from programmed cell death (PCD) events and termed hypersensitive response-like (HR-like) [18, 27, 28]. Within distinct cells–or preceding the development of HR-like lesions–higher levels of oxidative stress accelerate cell senescence (ACS response), with suites of structural symptoms closely mimicking those during autumnal senescence—another type of PCD [19, 29, 30]. However, the flux-based $O_3$ dose responsible for different types of cellular injuries and processes remains elusive. Furthermore, the dynamics of processes and structural injuries within cells are still poorly understood, missing the quantitative markers necessary for reproducible assessments, whereas the sequence of reactions appears to significantly determine the terminal symptom display [19]. The investigation of such processes is important for supporting the analysis of visible injury in foliage, a widely used indicator of $O_3$ stress in risk assessment [31–33] and monitoring studies [34, 35]. Furthermore, characterizing cellular responses can contribute to our understanding of oxidative stress traits, given the sometimes-confounding variability of symptoms, by directing our attention to symptom severity rather than visibility. Finally, such investigations can highlight the interplay with environmental conditions and specify the contribution of synergic environmental constraints to the display of visible symptoms [23, 24, 36].

The ontogenetic developmental stage of foliage also contributes to the $O_3$ stress response, as it is responsible for, e.g., characteristic gradients of visible symptoms within foliage in terms of distribution and frequency [37, 38]. A greater tolerance of younger *versus* older foliage at higher *versus* lower shoot position has been observed in the field and under controlled conditions, irrespective of the species and the environmental conditions [38–40]. For example, after eight days of exposure to 100 ppb $O_3$ ($POD_0 > 5$ mmol $O_3$ m$^{-2}$), Turc, Vollenweider [18] observed HR-like lesions on more than 6% percentage area in mature foliage *versus* less than 1%, in the case of expanding leaves. The mechanisms responsible for higher $O_3$ tolerance in younger foliage have remained rather elusive so far. Current evidence indicates that the observed tolerance is not caused by a lower $O_3$ uptake and dose [18, 39, 41]. The involvement of ROS during developmental processes such as cell expansion [42] or the senescence-like responses elicited by $O_3$ stress point to the significance of ontogenetic leaf development and ROS interactions regarding the dynamics of $O_3$ injury. However, $O_3$ injury assessments are performed almost exclusively on older foliage and evidence relating foliar ontogenetic development to the dynamics of $O_3$ injury is scarce.

After characterizing the early physiological and injury dynamics [18], the main objective of the present study was to deepen our understanding regarding the structural reactions to $O_3$ stress down to the cellular and subcellular level. Specifically, we analyzed the dynamics of structural and ultrastructural changes as a function of flux-based $O_3$ dose and leaf ontogenetic

development, testing the following hypotheses (H): (H1) structural injury develops in parallel with physiological and biochemical reactions; (H2) not only the structural markers but also their dynamics within mesophyll cells are response-specific; (H3) cellular reactions and ultra-structural markers of $O_3$ stress depend on ontogenetic leaf development; (H4) in the case of ACS responses and as a consequence of leaf ontogenetic development, time-based dynamics differ from flux-based dynamics. Response to $O_3$ stress were investigated in the framework of a replicated experiment with assessments in brightfield (LM), fluorescence (FM) and transmitted electron (TEM) microscopy. We used the same facility, plant material and exposure conditions as in Turc, Vollenweider [18] and we repeated the physiological and injury assessments completed in that study. Quantitative data used to characterize the dynamics of HR-like processes and ACS responses was obtained by means of a PCD assay [18] and biochemical measurements of chlorophyll content [43, 44]. In addition, chloroplast traits were assessed using TEM images and computer-assisted image analysis.

## Materials and methods

### Experimental setting

The experimental conditions, the tested material and the physiological and injury assessments are described in detail in Turc, Vollenweider [18]. Briefly, young hybrid poplar clones (*Populus tremula x alba*; INRAE 717-1b4) were micropropagated before growing for 1.5 months in a growth chamber (22˚C/18˚C day/night temperature and 75%/85% day/night relative humidity) with a 14 h photoperiod (Philips Son-T Agro 400 W lamps, PAR = 350 μmol m$^{-2}$ s$^{-1}$). The $O_3$ exposure was carried out using 48 poplar trees distributed across six identical phytotron chambers (PEPLor platform, Faculty of Sciences and Technologies, University of Lorraine) with three $O_3$ treatments (charcoal-filtered (CF) air, CF + 80 ppb $O_3$ and CF + 100 ppb $O_3$) and two replications of each (n = 16 trees). All foliar assessments were repeated on four trees per treatment at the 3$^{rd}$ and 10$^{th}$ leaf position from the tree base i.e., the youngest fully expanded leaf and the youngest leaf still expanding by the start of exposure. Stomatal conductance to water vapor ($g_w$) was measured 3 h after starting $O_3$ exposure using a LI-6400 portable photosynthesis system (LiCor, Inc, Lincoln, NE, United States) and inside the phytotron chambers). Measurements of $g_w$ were used to estimate cumulated $O_3$ uptake (POD$_0$; [18, 45].

### Assessment of ACS dynamics

The dynamics of degenerative changes indicative of ACS responses were assessed, measuring the foliar chlorophyll content [43, 44]. Leaf halves without midvein at the 3$^{rd}$ and 10$^{th}$ leaf position were repeatedly sampled (2, 8, 13, 23 days of treatment) and immediately shock-frozen in liquid nitrogen. They were homogenized in the same medium and the chlorophylls were extracted, solubilizing 100 mg of the homogenates in 1 ml acetone and incubating the mixture 15 min in the dark at 4˚C, prior to centrifugation (10 000 rpm / 10 min / 4˚C). The retrieved supernatants were diluted with distilled water [4:1, (v/v)] and absorbances were read at 663.2 nm and 646.8 nm (SAFAS UVmc1 spectrophotometer). For each sample, the extraction was repeated until the measured absorbance felt close to the detection limit. The estimates were converted into total chlorophyll concentrations (*Chl$_{tot}$*; mg g$^{-1}$ of FM) using the absorbance coefficients provided by Lichtenthaler [43]:

$$Chl_{tot} = 7.15 * A_{663.2} + 18.71 * A_{646.8}$$

The *Chl$_{tot}$* values for the sample aliquots were summed up.

## Assessments of cellular injury and HR-like dynamics

Dynamics of HR-like lesions were assessed by means of Trypan Blue vital staining and computer-assisted image analysis [18]. Therefore, two 6 mm discs per leaf position in four trees per treatment were sampled after 2, 8, 13 and 23 days of exposure. In addition, two other leaf discs were excised on each sampling date for structural and histochemical investigations of HR-like and ACS responses and dynamics at the cellular and subcellular level. The discs were fixed using EM-grade 2.5% glutaraldehyde buffered at pH 7.0 with 0.067 M Soerensen phosphate buffer, renewed after vacuum infiltration, and stored at 4˚C until further processing.

For LM and FM investigations, samples were dehydrated with 2-methoxyethanol (three changes), ethanol, n-propanol and n-butanol [46], then embedded in Technovit 7100 resin (Kulzer HistoTechnik). 1.5 µm semi-thin sections were trimmed from the center of the sampled discs using a Reichert Supercut 2050 microtome. Preparations were stained with Toluidine blue for observing the tissue and cell structure; callose deposits were specifically revealed using Aniline blue [47]. All assessments were performed using the 5× to 100× objectives of a Leica Leitz DM/RB microscope, diascopic or epifluorescent (UV excitation with filter cube A) light and an imaging system consisting of an INFINITY 2-1R camera and the Lumenera Infinity Analyze (release 6.4) software (Lumenera Corp., Ottawa, ON, Canada).

For TEM investigations, leaf blade patches excised from the center of EM-grade fixed samples were post-fixed in buffered 2% $OsO_4$, dehydrated by successive baths of graded ethanol, infiltrated by a series of graded propylene oxide/Epon 812 mixtures (with DDSA, NMA, DMP hardener), and embedded in Epon. Ultra-thin sections (70 nm) were cut using a Reichert UltraCut S ultra-microtome, mounted on gold grids, and contrasted using saturated uranyl acetate in 50% ethanol and lead citrate (Reynold procedure). Sections were observed using a Philipps CM12 transmission electron microscope.

The dynamics of ACS-related injury were assessed by means of chloroplast trait measurements, as this organelle is particularly sensitive to oxidative stress and shows characteristic alterations during senescence processes [19, 48, 49]. Using TEM pictures (17500× magnification) and the ImageJ software (vers. 2.0.0; [50]), size (area/perimeter/long and short axis/area of plastoglobuli and starch grains) were assessed by applying computer-assisted image analysis. Chloroplast shape was estimated by calculating chloroplast circularity ($Cir_{chloro}$), following the formula:

$$Cir_{chloro} = \frac{A_{chloro} * 4\pi}{P_{chloro}{}^2}$$

where $A_{chloro}$ and $P_{chloro}$ are the chloroplast area and perimeter, respectively.

Measurements were repeated, selecting ten cross-sectioned chloroplasts within 3 to 4 cells per upper and lower palisade parenchyma layer within each sample. Together with chlorophyll data, the measured chloroplast traits were used to calculate conceptual models synthesizing the ultrastructural responses in chloroplasts in response to experimental factors. Elliptical representations of chloroplasts, accumulated starch grains and plastoglobuli percentage areas were drawn using the *Plotrix* package [51] of R software, version 3.5.0 [52].

## Statistical analysis

The dynamics of injury, ultrastructural and physiological responses to experimental factors were analyzed using linear mixed effects (LME) models. The fixed-effect factors included the time or $POD_0$, O₃ treatment, leaf position, mesophyll layer (chloroplast ultrastructural data), and their interactions. The random-effect factors included the tree nested in the chamber (leaf data, with leaf position forming the statistical unit) and the cell nested in the tree (chloroplast

ultrastructural data, with chloroplast forming the statistical unit). All statistical analyses (LME models and post-hoc tests) were performed using the *lme4* [53] and *lsmeans* [54] R packages.

## Results

### Dynamics of leaf responses

In the 100 ppb $O_3$ treatment at both leaf positions (Fig 1, S1 Fig), the development of lesions (especially non-oxidized) showed a clear sigmoid-like dynamic. Despite rather synchronous dynamics, the leaf percentage area showing non-oxidized *versus* oxidized HR-like lesions was always greater, with the exceedance reaching an 8.4× order of magnitude at both leaf positions in the 100 ppb $O_3$ treatment by the end of the experiment (Fig 1 and S1 Fig; oxidation state: $P < 0.001$; oxidation state*time: $P < 0.001$). Differences between the 100 and 80 ppb $O_3$ treatment in the leaf percentage area showing non-oxidized and oxidized lesions at the two leaf positions became significant after 13 days of exposure (Fig 1 and S1 Fig). Similar but weaker injury and a delayed onset of HR-like processes were observed at the 10th *versus* the 3rd leaf position (S1 Fig *versus* Fig 1; $O_3$ treatment*leaf position: $P < 0.05$, $O_3$ treatment*time*leaf position: $P < 0.001$).

Also, in the case of chlorophyll measurements, the $O_3$ effects were generally more pronounced in the 100 *versus* 80 ppb $O_3$ treatment, with the chlorophyll decrease triggered by $O_3$ stress observed after 13 days of exposure (Fig 2A), and the $O_3$ treatment factor being then significant ($O_3$ treatment: $P < 0.001$, $O_3$ treatment*time: $P < 0.001$). Reported to the chlorophyll amounts in the CF treatment, the relative chlorophyll content in trees from the two $O_3$ treatments was also monotonically decreased with increasing $POD_0$ (Fig 2B; $POD_0$: $P < 0.001$). However, the other effects were non-significant, probably as a consequence of lower stomatal conductance ($g_w$) and an earlier $g_w$ drop in the 100 ppb $O_3$ treatment (S2A Fig) causing low $POD_0$ differences between the two treatments (S2B Fig).

### Structural evolution of HR-like lesions proceeded independent of $O_3$ exposure or leaf position

Dead cells within HR-like lesions showed *post-mortem* evolution of their cellular ultrastructure. Using leaf cross-sections *versus* disc samples (Fig 3E and 3F *versus* 1B inset), HR-like lesions in the CF + 100 ppb $O_3$ treatment were detected after 13 *versus* 8 days of exposure, once the leaf percentage area of lesions at both leaf positions was close to 10% (Table 1, Fig 1 and S1 Fig). The first HR-like lesions were observed in the lower palisade parenchyma. At an early developmental stage (stage 1), cell death was indicated by cell cytorrhysis, in apparent causal link with vacuole collapse, segmentation and tonoplast rupture (Fig 3E and 3F). This initiation of cell content disruption occurred before any alteration evidence in the surrounding organelle structure could be observed. At an intermediate stage of HR-like reaction development (stage 2), the organelles became increasingly disrupted, the cytoplasm condensed, and the cell integrity was lost (Fig 3G–3K'). Several hallmarks of oxidative stress and HR-like reactions (Table 1) became apparent, such as (1) degradation of the mitochondrial matrix and envelope (Fig 3H), (2) nucleus pyknosis (Fig 3I), (3) chloroplast disruption (Fig 3J), and (4) leakage of cellular debris into the apoplast through cell wall breaks (Fig 3K and 3K'). In the final stage of HR-like reaction development (stage 3), the cell content appeared completely disrupted, with amorphous remnants–except starch grains–densely fused together (Fig 3L–3N). The surrounding living cells showed local cell wall thickening, following callose deposition on the cell wall portion adjacent to dead HR-like cells (Fig 3O and 3P). These three stages of HR-like reaction development and their markers (Table 1) could be

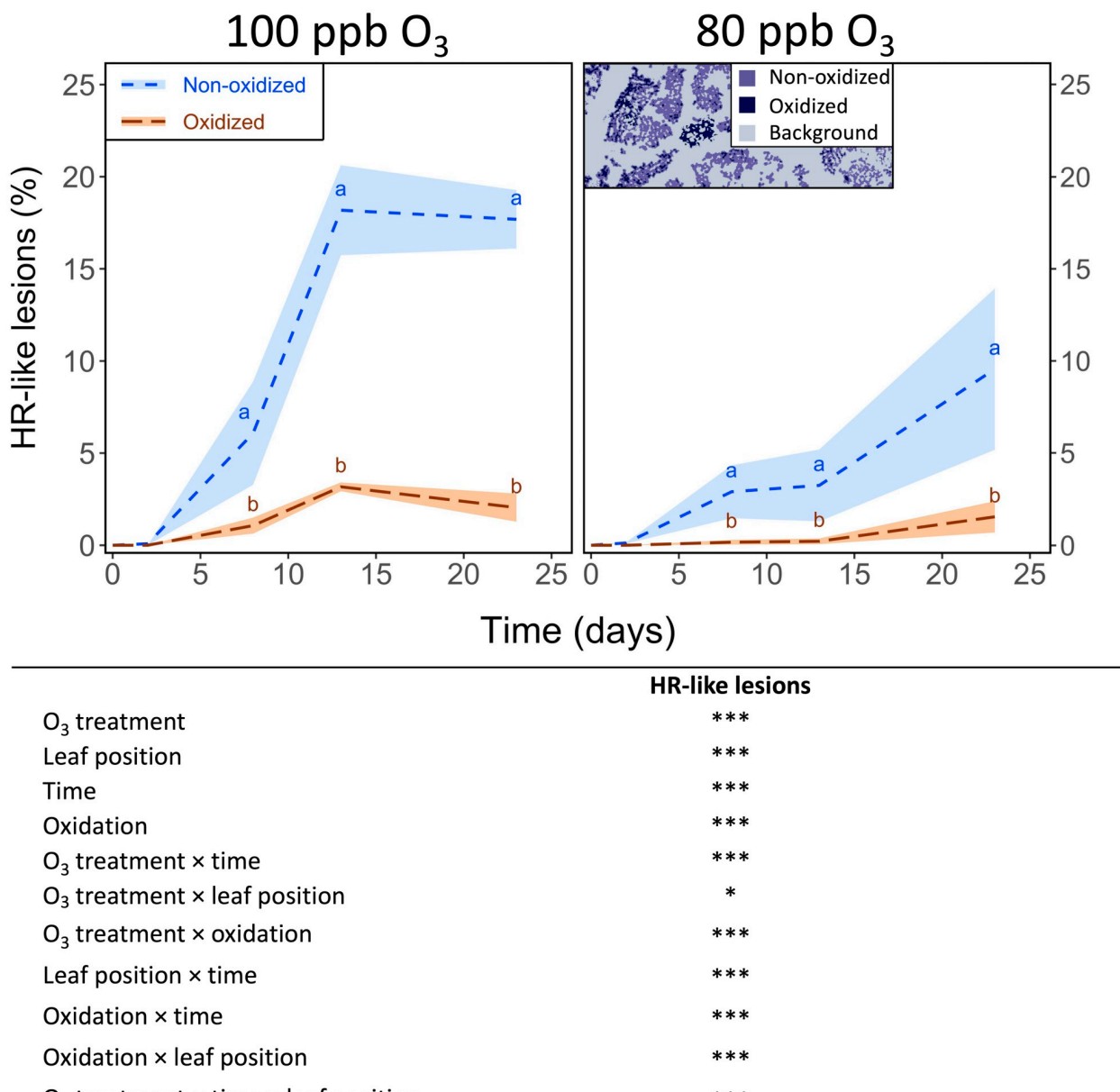

**Fig 1. Development dynamics of non-oxidized and oxidized HR-like lesions at the 3rd leaf position in hybrid poplar (*Populus tremula x alba*), as a function of $O_3$ treatment and time.** {model: lmer[log(variable+1) ~ leaf position * Oxidation * Time * $O_3$ treatment + (1|pot)]; *** $P \leq 0.001$; * $P \leq 0.05$}. The inset image is a synthetic image of the particle distribution and morphology in each lesion color class (non-oxidized/oxidized) during image analyses of HR-like reactions. Values represent percentage area means ± SE of leaf discs showing non-oxidized or oxidized HR-like lesions (n = 4). Different letters indicate significant differences between treatments at a given assessment date (Tukey's honestly significant difference (HSD) post-hoc test, $P \leq 0.05$).

observed within adjacent cells or cell groups scattered in the lower or upper palisade parenchyma layer. The HR-like stages and cell ultrastructure showed similar characteristics, irrespective of $O_3$ treatment, leaf position and exposure duration (13, 23 days), with apparent *post-mortem* evolution.

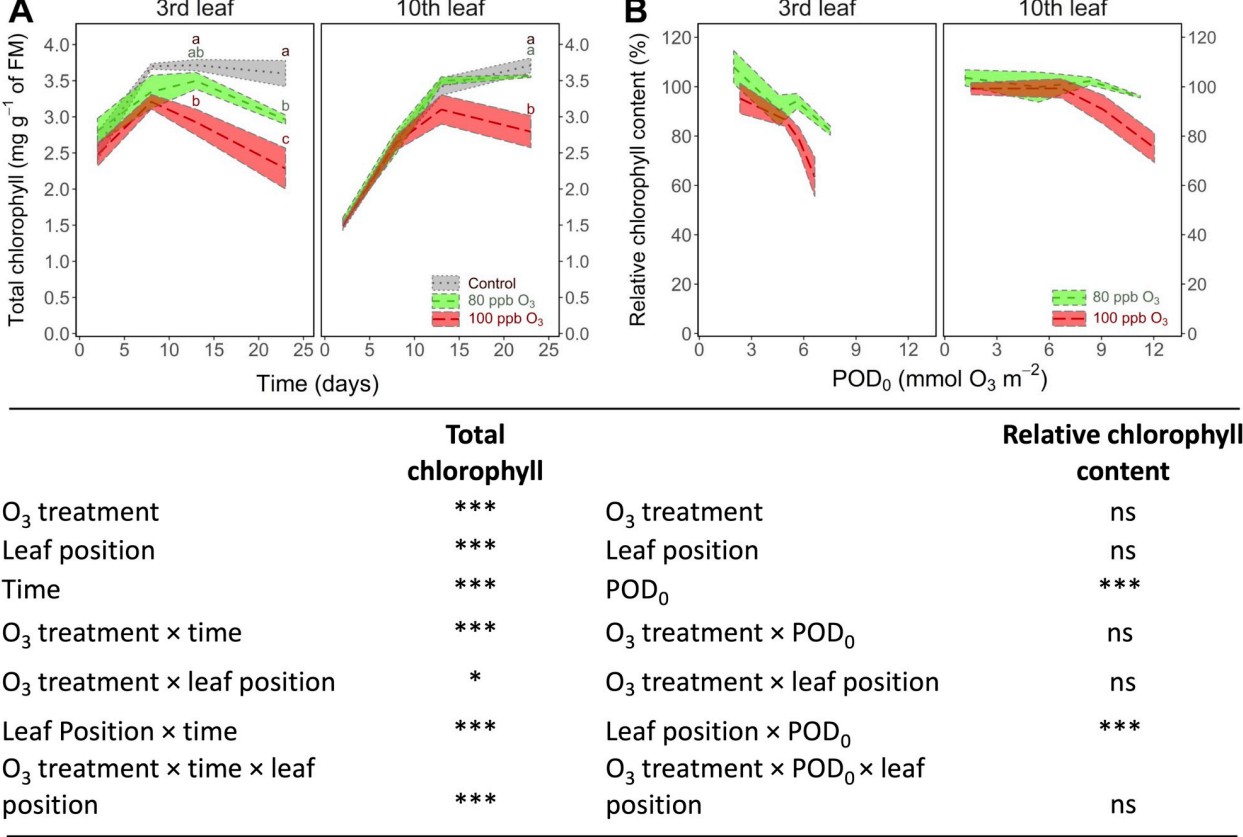

**Fig 2. Dynamics of changes in the total chlorophyll concentration within leaves of hybrid poplar (*Populus tremula x alba*) as a function of assessment time (A; mg g$^{-1}$ of fresh matter), POD$_0$ (B; percentage of values in the CF treatment).** $O_3$ treatment (dotted line: CF +80 ppb $O_3$, long dash line: CF +100 ppb $O_3$), leaf position, and their interactions (model: lmer(variable ~ $O_3$ treatment * leaf position * time or POD$_0$ + (1 | tree/chamber)); *** $P \leq 0.001$, * $P \leq 0.05$, ns not significantly different). Values represent means ± SE (n = 4). Different letters indicate significant differences between treatments on a given assessment date (post-hoc Tukey's HSD, $P \leq 0.05$).

## Ultrastructural traits showed degenerative dynamics at least partly determined by $O_3$ exposure and leaf position

Mesophyll cells surrounding the HR-like lesions showed typical structural and ultrastructural markers of ACS in response to the $O_3$ treatments. These injuries represented an obvious aggravation and acceleration of ontological senescence processes observed in CF material of comparable age (Fig 4K–4O and 4P–4T *versus* Fig 4A–4E and 4F–4J; Table 1). At the cellular level, the injuries included a relative increase in vacuome size at the expense of cytoplasm and an apparent reduction in the number and size of chloroplasts (Fig 4K, 4L, 4P and 4Q *versus* Fig 4A, 4B, 4F and 4G). At the subcellular level, we observed (1) cell wall thickening (Fig 5), (2), various degenerative changes (poor thylakoid membrane resolution following oxidative injury (Fig 4R and 4T *versus* Fig 4M and 4O); increase in plastoglobuli size and frequency (Fig 4G, 4H, 4Q and 4R, versus Fig 4B, 4C, 4L and 4M)) and larger starch grains in chloroplasts (Fig 4K–4M and 4P–4R *versus* Fig 4A–4C and 4F–4H), (3) missing nucleoli indicative of lower nuclear transcription activity (Fig 4N and 4S *versus* Fig 4D and 4I), and (3) accumulation of secondary metabolites in the vacuole (Fig 4P *versus* Fig 4F). The ACS-related changes became apparent after 13 days of treatment (Table 1). Until the end of the experiment, they appeared more severe in the 100 ppb $O_3$ treatment and at the 3$^{rd}$ leaf position. Cell wall thickening processes were specific to leaf position: at

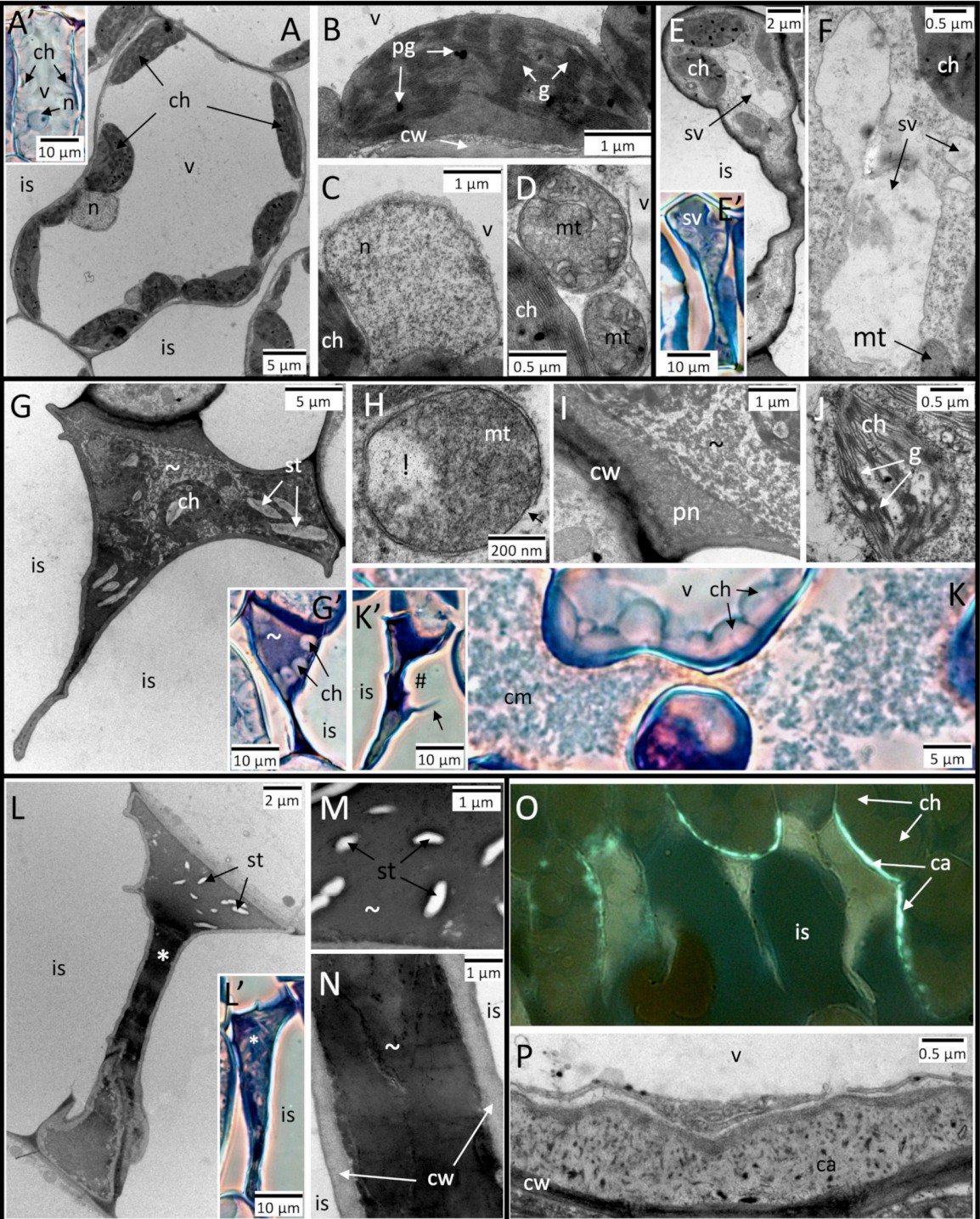

**Fig 3.** Evolution of cellular and subcellular traits in HR-like lesions within mesophyll cells of hybrid poplar (*Populus tremula x alba*) leaves at the 3rd (A–D, K–P) and 10th leaf position (E–J) and in response to the CF+100 ppb O₃ treatment. **A–D** Asymptomatic mesophyll cell structure in the CF treatment. **E–F** Early stage of HR-like reaction development: cell death and cytorrhysis. In apparent relation to cell cytorrhysis, the vacuole was shrunken (sv) and its irregular shape indicated tonoplast rupture, whereas other organelles were still intact. **G–K** Intermediate stage of HR-like reaction development: disruption of cell content. **H** Within mitochondria, degradation of the mitochondrial matrix (!) and disruption of outer membrane (arrow). **I** Nucleus pyknosis (pn) and condensation of cytoplasm (~). **J** chloroplast (ch) disruption. **K–K'** Leakage of cell debris (cm) into the intercellular space (is) through break points in the cell wall (#). **L–N** Final stage of HR-like reaction development: condensation and fusion of cellular debris. **O–P** Local thickening of cell wall (cw) through callose deposition within cells adjacent to lesions. Other structures: g: grana, n: nucleus, v: vacuole. Technical specifications: post-fixation

using OsO4; contrasting using uranyl acetate and lead citrate; observation in TEM (**A, B–E, F, G, H-J, L, M, N, P**); staining with Toluidine blue and observation under phase contrast in bright field microscopy (**A', E', G'-K', L'**); staining with Aniline blue and observation with fluorescence microscopy (**O**).

the 3rd leaf position, multivesicular bodies discharged degraded cellular remnants into the periplasm by exocytosis, while little physiological activity could be observed in the cytoplasm (Fig 5A–5C, Table 1). At the 10th leaf position, dense networks within the cytoplasm of (1) burgeoning rough endoplasmic reticulum, (2) dictyosomes, with their trans domains and the secreted vesicles oriented towards cell walls, and (3) numerous mitochondria all indicated ontological cell wall thickening processes, together with cellulose microfibril deposition (Fig 5D–5G, Table 1).

Chloroplast traits were responsive to changes in the O₃ exposure and showed a clear evolution over time, with significant differences between the mesophyll layers. Primarily the

**Table 1. Development of structural and ultrastructural markers of O₃ stress over 30 days of exposure to the CF +100 ppb O₃ and CF +80 ppb O₃ treatment *versus* CF treatment, within the mesophyll of hybrid poplar leaves (*Populus x tremula alba*).**

| Compartment | Structural marker | Physiological process | Experimental factors | | | | | | | | | Fig. |
|---|---|---|---|---|---|---|---|---|---|---|---|---|
| | | | Treatment | 100 ppb O₃ | | | | | | 80 ppb O₃ | | |
| | | | Leaf position | 3rd | | | 10th | | | 3rd | 10th | |
| | | | Exposure (*days*) | 8 | 13 | 23 | 8 | 13 | 23 | 13 | 13 | |
| Leaf blade | Leakage of cellular material into apoplast | HR-like | | + | - | - | - | + | - | - | - | 3K |
| Cell | Cytorrhysis | HR-like | | - | + | + | - | + | + | + | - | 3G and 3L |
| | *Rupture* of cell wall | HR-like | | - | - | + | + | - | + | + | - | 3K' |
| | *Disruption of cell content* | | | | | | | | | | | |
| | Rupture of membrane (plasmalemma and tonoplast) | HR-like | | - | + | + | - | + | + | + | - | 3E–3F |
| | Disruption of organelles | HR-like | | - | + | + | - | + | + | - | - | 3H and 3J |
| | Condensation of cellular debris | HR-like | | - | - | + | + | - | + | + | + | 3L–3N |
| | *Cell wall thickening* | | | | | | | | | | | |
| | Wart-like protrusion | Oxidative stress | | - | - | + | - | - | + | - | - | Not shown |
| | Discharge of autophagic vacuoles | ACS | | - | - | + | - | - | - | - | - | 5A |
| | Cell wall synthesis | Leaf ontology | | - | + | + | + | + | + | - | - | 5D–5G, |
| | Callose deposition | HR-like | | - | + | + | - | - | + | - | - | 3O and 3P |
| Cytoplasm | Autophagy | ACS | | - | - | + | - | - | - | - | - | 5B and 5C |
| | Condensation | ACS | | - | + | + | - | + | + | - | - | not shown |
| Vacuole | Size increase | ACS | | - | + | + | - | | + | + | - | 4Q vs 4G |
| | Collapse | HR-like | | - | + | + | - | + | + | + | - | 3F |
| | Accumulation of secondary product in the vacuole | Oxidative stress ACS | | - | + | + | - | - | + | - | - | 4P |
| Nucleus | Picnosis | HR-like | | - | + | - | - | - | - | - | - | 3I |
| | Condensation | ACS | | | - | + | - | - | + | - | - | 4S |
| Chloroplast | Size decrease | ACS | | - | + | + | - | - | + | - | - | 6 |
| | Increase in size and density of plastoglobuli | ACS | | - | + | + | - | + | + | + | - | 4R and 4T vs 4M and 4O |
| | Higher circularity | ACS | | - | - | + | - | - | + | - | - | 6 |
| | Disruption | HR-like | | - | + | + | - | + | + | + | - | 3J |
| Mitochondria | Disruption | Oxidative stress | | - | + | + | - | + | + | + | + | 3H |

"+" and "-" indicates presence and absence of marker, respectively.

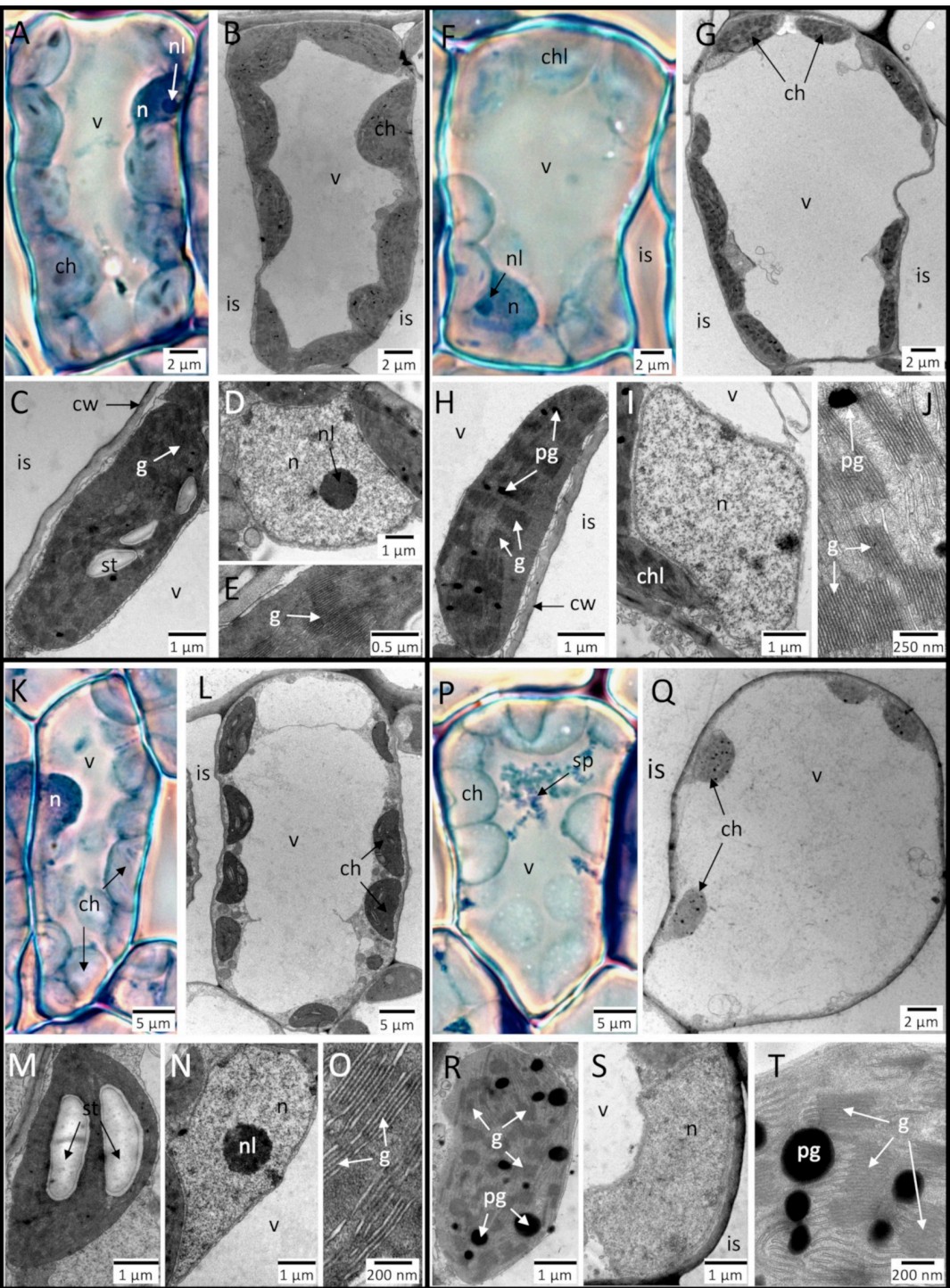

**Fig 4.** Evolution of degenerative traits at the cellular and subcellular level within mesophyll cells surrounding HR-like lesions inside of hybrid poplar (*Populus tremula x alba*) leaves at the 3rd leaf position and in response to the CF+100 ppb O₃ (K–O, P–T) treatment relative to the CF treatment (A–E, F–J). Treatment time: 8 days (**A–E, K–O**) or 23 days (**F–J, P–T**). The main difference between the two treatments was the quicker evolution of degenerative traits in the O₃-treated trees (**F, G**) relative to the CF trees (**P, Q**). Markers at the cellular level: relative increase in vacuole (v) size at the expense of cytoplasm and the apparent reduction in the number and size of chloroplasts (ch). Markers at the subcellular level: within chloroplasts: increase in starch grain (st) accumulation (**K–M** *vs.* **A–C**), injury to thylakoid membranes of grana (g; **R, T** *vs.* **H, J**) and increase in the plastoglobuli (pg) size and frequency (**Q, R,** *vs.* **G, H**); within nucleus (n): condensation of chromatin and loss of nucleoli (nl; **N, S** *vs.* **D, I**); in the vacuole: accumulation of secondary metabolites (sp; **P** *vs.* **F**). Other structures: cw: cell wall, is: intercellular space. Technical specifications: post-fixation using OsO₄, contrasting using uranyl acetate and lead citrate, observation in TEM (**B–E, G–J, L–O, Q–T**); staining with Toluidine blue, observation under phase contrast in bright field microscopy (**A, F, K, P**).

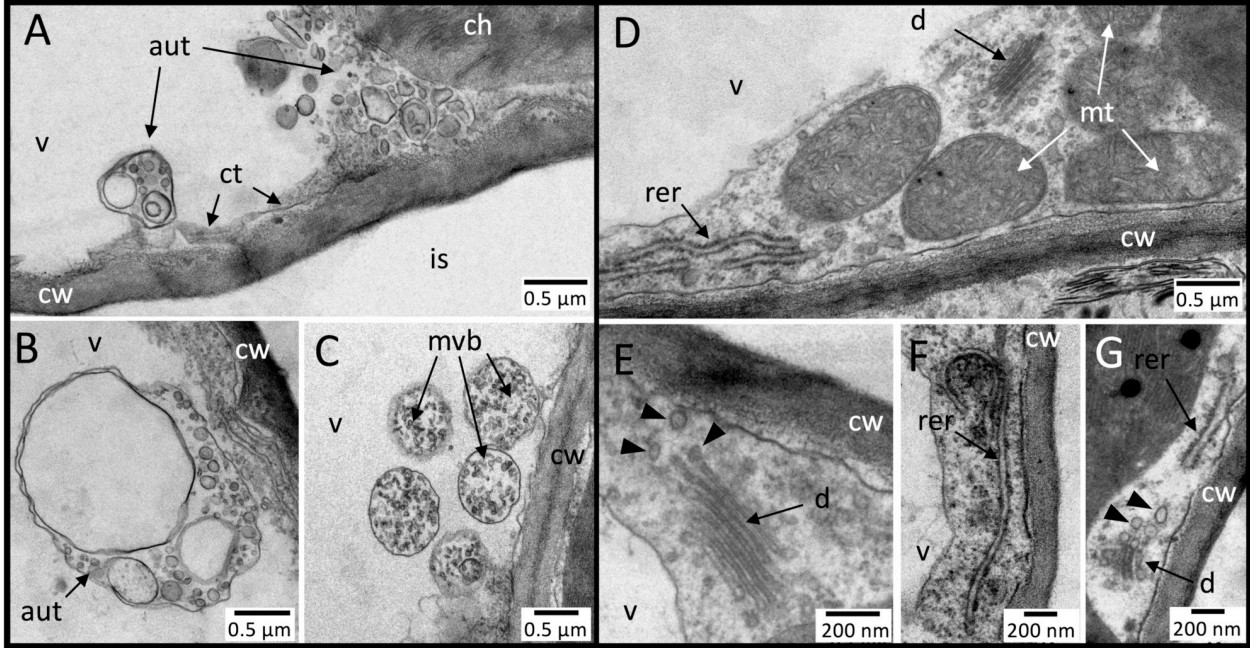

**Fig 5. Contrast in the cell wall thickening processes at the 3$^{rd}$ (A–C) *vs.* 10$^{th}$ leaf position (D–G) within leaf mesophyll cells undergoing ACS processes from hybrid poplar (*Populus tremula x alba*) trees in the CF+100 ppb O$_3$ treatment. A–C** At the 3$^{rd}$ leaf position, cell wall thickening primarily resulted from exocytosis (~) of multivesicular body (mvb) content into the periplasm (pe), as a consequence of autophagic processes. Notice the thin cytoplasm (ct) layer and few organelles. **D–G** At the 10$^{th}$ leaf position, ontological cell wall thickening in response to treatment was indicated by a dense network of mitochondria (mt; **D**), rough endoplasmic reticulum (rer; **F**), and dictyosomes (d; **E**) secreting vesicles (arrowheads in **E**, **G**) heading towards apoplast. Other structures: ch: chloroplast, cw: cell wall, is: intercellular space, v: vacuole. Technical specifications: post-fixation using OsO$_4$, contrasting using uranyl acetate and lead citrate, observation in TEM.

chloroplast shape (circularity) and the total plastoglobuli area were affected by the O$_3$ treatment and increasing POD$_0$ (Fig 6; $P < 0.001$). By the end of the exposure in the 100 ppb O$_3$ *versus* CF treatment, the plastoglobuli area was 10-fold larger and the chloroplast circularity had increased by 40%, irrespective of leaf position (Fig 6, S1 Table; O$_3$ treatment*time: $P < 0.001$). As indicated by significant effects of leaf position ($P < 0.001$) and time ($P < 0.001$, plastoglobuli only), these two parameters also responded to ontological developmental processes, which O$_3$ exposure accelerated (O$_3$ treatment or POD$_0$*leaf position: $P < 0.05$). Chloroplasts in the lower palisade parenchyma layer tended to be rounder (Fig 6, S1 Table; mesophyll layer: $P < 0.001$). Chloroplast size decreased over time ($P < 0.001$), most likely corresponding to foliar phenological development, but it was not responsive to O$_3$ treatment and POD$_0$. Starch grain size and abundance responded to apparent chloroplast illumination and foliar phenology, as indicated by significant effects of leaf position, time, mesophyll layer and their interactions (Fig 6; $P < 0.01$). Comparing the 100 *versus* 80 ppb O$_3$ treatment on one assessment date (13 days of O$_3$ exposure), the findings appeared rather similar and the sensitivity of plastoglobuli to oxidative stress was further confirmed (S3 Fig; O$_3$ treatment and POD$_0$: $P < 0.001$).

## Discussion

### Contrasts between HR-like and ACS dynamics

In confirmation of Turc, Vollenweider [18], sigmoid-like and monotonic plant response dynamics appeared to be characteristic of HR-like and ACS processes within foliage under O$_3$

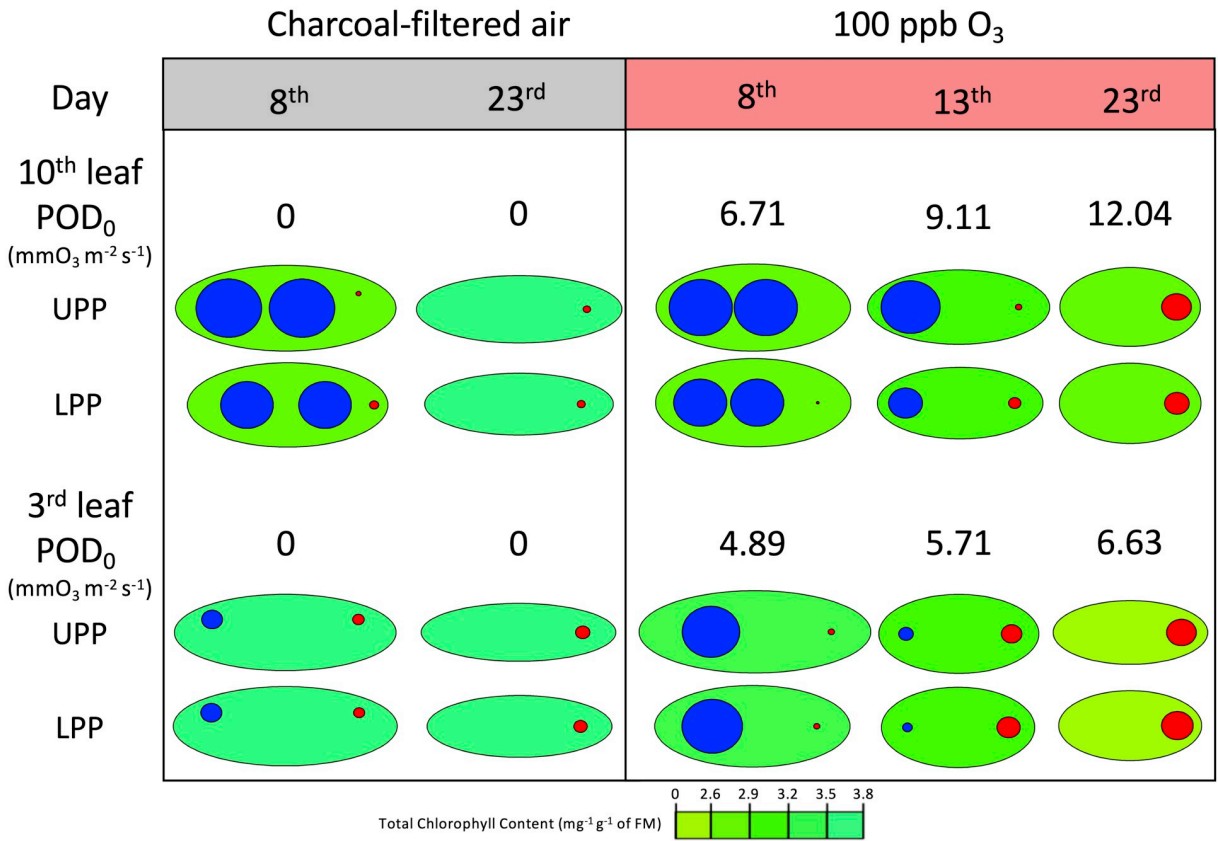

| | Chloroplast size | Chloroplast circularity | Starch grain area | Total plastoglobuli area | | Relative chloroplast Size | Relative chloroplast circularity | Relative grain starch area | Relative total plastoglobuli area |
|---|---|---|---|---|---|---|---|---|---|
| $O_3$ treatment | ns | *** | ns | *** | $O_3$ treatment | ns | *** | ns | *** |
| Leaf position | ns | *** | *** | *** | Leaf position | ns | ** | ns | ns |
| Time | *** | ns | *** | *** | $POD_0$ | ns | *** | ns | *** |
| Mesophyll layer | ns | *** | *** | ns | Mesophyll layer | ns | ns | ns | ns |
| $O_3$ treatment × time | ns | *** | ns | *** | $O_3$ treatment × $POD_0$ | ns | ns | ns | ns |
| $O_3$ treatment × leaf position | ns | * | ns | *** | $O_3$ treatment × leaf position | ns | ns | ns | ns |
| Leaf position × time | ns | ns | ** | ns | Leaf position × $POD_0$ | ns | ** | ns | ** |
| Leaf position × mesophyll layer | ns | ** | *** | ns | Mesophyll layer × leaf position | ns | ns | ns | ns |
| Mesophyll layer × time | ns | ns | *** | ns | Mesophyll layer × $POD_0$ | ns | ns | ns | ns |

**Fig 6. Visualization of chloroplast size and shape trait dynamics as a function of $O_3$ treatment, leaf position, assessment time or $POD_0$, mesophyll layer, and interactions, as observed in the leaf mesophyll of hybrid poplar trees (*Populus tremula x alba*) on the 8th, 13th and 23rd day of treatment.** Model: lmer(variable ~ $O_3$ treatment * leaf position * time + (1 | cell/tree)); *** $P \leq 0.001$, * $P \leq 0.05$, ns not significantly different). Data basis for the conceptual models of chloroplast trait changes: chloroplast long and short axis (green ellipse), cumulated starch grain and plastoglobuli percentage area (blue and red circles), total chlorophyll content (in mg g$^{-1}$ FM; green color scale).

stress. In this replicated experiment, the HR-like reactions and their dynamics at the different leaf positions matched quite well with those previously observed [18]. The duration of the latency period and plateau values were strikingly similar in the two experiments, irrespective of treatment and leaf position. However, the beginning of degenerative reactions was different when the total chlorophyll content was assessed rather than the reflectance. Although reflectance-based has been shown to be one of the best ways to measure chlorophyll content without destroying plants [55, 56], in our study the assessment method appeared important, with the biochemical assay being more accurate and specific to chlorophylls than reflectance-based estimates. Nevertheless, the dynamics of the ACS plant response appeared little affected by the assessment method, with the results from this replicated experiment overall matching quite well those obtained previously.

While rather monotonic degenerative responses to O₃ stress have been observed in various studies [23, 57, 58], experimental confirmation of the sigmoid-like dynamics in the case of HR-like lesions has been missing so far, instead remaining an implicit hypothesis based on apoptosis science [59, 60]. As cell death is the earliest event during HR-like lesion development, its selection as the structural marker being monitored allowed us to sensitively and comprehensively assess HR-like events and dynamics. The numerous other, more specific apoptosis markers were found to appear afterwards, during the *post-mortem* ontological development of the lesion. This is the first time, to our knowledge, that structural injury resulting from HR-like processes has been shown to develop in dead cells and thus without biological control. Some of the released cell debris–notably from cell walls–may act as signals within the surrounding tissues [61, 62]. The three observed stages of HR-like reaction development may correspond to an injury severity scale or result from successive degradation of dead cell contents. Arguments in favor of the former interpretation include (1) the occurrence of all three stages during the first observation of HR-like reactions and (2) the occurrence of all three stages within the same cell in some cases, with part of the cell still in disruption and the rest with fused remnants (*i.e.*, Fig 3G). The argument that the stages of HR-like reaction development are successive primarily rely on the concept of delayed lesion oxidation, which increases over time [18]. The non-oxidized HR-like lesions appeared to correspond to cells in the first stage of HR-like reaction development, as organelles were not disrupted but rather intact, as indicated by the presence of chlorophyll [18]. Since oxidized lesions had a disrupted cell content, the cellular material in stage 3 and at least part of that in stage 2 should be included in the oxidized lesion class during quantitative assessments.

Callose deposits on cell walls as a consequence of O₃ exposure form a typical HR-like marker [21, 36, 63], which was observed within mesophyll cells adjacent to lesions in poplar leaves. They are also found in the case of plant–pathogen interactions and hypersensitive responses (HR), strengthening cell walls and slowing pathogen-induced damage [64]. Callose deposition is promoted by salicylic acid and suppressed by jasmonic acid [65], and the observed deposition pattern may result from interacting effects of both hormones [66].

Whilst occurring at an early stage of HR-like injury, ACS represented a terminal programmed cell death (PCD) event, which had not yet been completed in material analyzed in LM and TEM. Dynamics of ACS-related changes proceeded as a coordinated decrease in apparent cellular activity and chlorophyll content concomitant with various subcellular degenerative changes, especially inside of chloroplast. Similar traits were observed during autumnal senescence in other species, such as rice [67], Scots pine [68] and beech [48]. In the case of beech, plastoglobuli were larger and displayed more intense extrusion to the vacuole than in poplar. It suggests a larger degree of lipid remobilization processes in the former species, which might be associated with beech competitive and shade-tolerance traits [69].

Two chloroplast traits, namely circularity and total plastoglobuli area, were particularly responsive to $O_3$ treatment and $POD_0$. They also showed alteration with leaf aging and thus responded in a way similar to total chlorophyll content. Taken together, the measured traits indicated injury to thylakoids and pointed the chloroplast inner membrane system as being particularly sensitive to oxidative stress [70]. During senescence, decreased antioxidant capacities in chloroplasts trigger ROS production, resulting in a feedback loop accelerating the senescence process [71]. $O_3$ stress, and high levels of photo-oxidative stress observed in the field [19], may contribute to further accelerating the degenerative processes. The chloroplast size and starch grain accumulation responded to leaf aging and chloroplast illumination only, while they were not affected by $O_3$ treatment or dose.

### Leaf-age-mediated responses to $O_3$ stress

A reduction in total chlorophyll content, emergence of HR-like lesions, and ACS-related microscopic changes occurred later and with a larger $POD_0$ at the 10th versus 3rd leaf position, consistent with a higher tolerance of developing leaves [37, 39, 72]. Being observed at both leaf positions, the cell wall thickening activity proceeded in a different way, depending on the ontological development reached at each foliage level. At the lower leaf position, cell wall thickening was promoted by autophagic processes, in direct association with the degeneration of cellular content. The tolerance gained thus appears uncertain, and such processes may even contribute to accelerated leaf shedding, as observed during experimental exposure of mature non-acclimated foliage to O3 [23, 73]. In top leaves, cell wall thickening proceeded as during cell growth. This latter process is adaptive [74] and implies tight transcriptional regulation of ROS homeostasis [42, 75], which may help in coping with additional $O_3$-induced ROS. The two thickening processes thus appeared to be functionally related to lower (3rd leaf position) versus enhanced (10th leaf position) $O_3$ tolerance. They are consistent with observed increase in stress susceptibility during sink-to-source transition in maturing foliage [76]. Further supporting this idea, the ACS-related changes in $O_3$-exposed foliage of tested poplar trees started once the sink-to-source transition had apparently been achieved, as indicated by the plateau reached in chlorophyll content and stomatal conductance within the CF trees.

### Leaf response and visible injury dynamics

Macroscopic injury, used in risk assessment studies [31–33], represents a terminal reaction resulting from different types of structural injury, with specific dynamics and plant responses proceeding at different paces. With only the oxidized HR-like lesions contributing to stippling-like visible injury [18], a significant fraction of disruptive processes may remain undetected, at least at an early stage of symptom development. As a consequence, the rapidity and extent of structural responses to $O_3$ stress appear to be generally underestimated. Cell death detection in less controlled experimental conditions may lack specificity. ACS-related changes, which responded sensitively to increasing $POD_0$ in our study, have a similar limitation. In multi-stress situations, quantitative or observational assessments should therefore rely on a combination of markers. As assessments are generally completed late during the vegetation season and rarely repeated, the observed injury should be considered the cumulative result of multiple events, depending on each response occurring before the sampling date [19]. This finding may provide the principal explanation for the spatial and temporal variability in visible injury observed in field conditions.

## Conclusion

In this study, we investigated the dynamics of structural and ultrastructural changes occurring in poplar leaves in response to two $O_3$ levels and as a function of time, phytotoxic $O_3$ dose ($POD_0$), leaf developmental stage, and mesophyll layer. Although the development dynamic of HR-like lesions depended on $O_3$ concentration and $POD_0$, most of the HR-like structural and ultrastructural markers appeared after cell death, independent of the experimental factors. By contrast, the pace of degenerative ACS processes was closely depending on the $O_3$ levels, $POD_0$ and leaf development. At the subcellular level, thylakoid membranes in chloroplasts were identified as being one of the most sensitive subcellular structures to $O_3$ stress. Concerning our experimental hypotheses (H), structural injury developed in parallel with physiological and biochemical reactions in the case of ACS but not HR-like reactions (partial rejection of H1). Dynamics of HR-like and ACS responses, as well as structural markers, were response-specific (confirmation of H2), with cellular reactions being the same at the 3rd and 10th leaf positions, except cell wall thickening processes (partial confirmation of H3). Finally, dynamics of ACS responses in leaves with different leaf ontological progress differed over time and with $POD_0$ (confirmation of H4). With specific markers and contrasted dynamics, ACS and HR-like processes appear fairly independent, which provide a rationale for observed variability in the symptom display. A combination of easily assessed ACS and HR-like markers, such as chlorophyll reflectance and mesophyll cell death, appears best suited for specifically quantifying $O_3$ effects in trees during *e.g.* risk assessment studies. In a time of rapid climate change and given the varying interplay of HR-like and ACS processes with environmental conditions, an open question is how foliar responses to $O_3$ stress may evolve in the future.

## Supporting information

**S1 Fig. Development dynamics of non-oxidized and oxidized HR-like lesions at the 10th leaf position in hybrid poplar (*Populus tremula x alba*) as a function of $O_3$ treatment and time.** {model: lmer[log(variable+1) ~ leaf position * oxidation * time * $O_3$ treatment + (1|pot)]; *** $P \leq 0.001$; * $P \leq 0.05$}.}. The inset image is a synthetic image of the particle distribution and morphology in each lesion color class (non-oxidized/oxidized) during image analyses of HR-like reactions. Values represent percentage area means ± SE of leaf discs showing non-oxidized or oxidized HR-like lesions (*n* = 4). Different letters indicate significant differences between treatments at a given assessment date (Tukey's honestly significant difference post-hoc test, $P \leq 0.05$).
(TIFF)

**S2 Fig. Dynamics of changes in the (A) stomatal conductance to water ($g_w$) and (B) phytotoxic $O_3$ dose ($POD_0$) of hybrid poplar leaves (*Populus tremula x alba*) as a function of $O_3$ treatment (CF ●, CF+80 ppb $O_3$ ▲, CF+100 ppb $O_3$ ■), leaf position and time of assessment.** {model: lmer[sqrt(variable)] ~ $O_3$ treatment * leaf position * time + (1 | tree/chamber); ***$P \leq 0.001$}. Values represent means ± SE (n = 4). Different letters indicate significant differences between treatments for a given assessment date (Tukey's honestly significant difference post-hoc test, $P \leq 0.05$).
(TIFF)

**S3 Fig. Dynamics of size and shape changes in chloroplast traits as a function of $O_3$ treatment, leaf position, $POD_0$, mesophyll layer and interactions, as observed in the leaf mesophyll of hybrid poplar trees (*Populus tremula x alba*) on the 13th day of treatment.** Model: lmer(variable ~ $O_3$ treatment * leaf position * $POD_0$ + (1 | cell/tree)); *** $P \leq 0.001$, * $P \leq 0.05$, ns not significantly different). Green ellipse: chloroplast, blue circle: total starch grain area, red

circle: total plastoglobuli area. Green shading represents the total chlorophyll content (mg g$^{-1}$ of FM).
(TIFF)

**S1 Table. Evolution of chloroplast size and shape in the leaf mesophyll of hybrid poplar trees (*Populus tremula* x *alba*) in response to O₃ treatment, leaf position, mesophyll layer and assessment time.** Values represent mean ± SE, n = 4. Different letters indicate significant differences between treatments for a given assessment date (Tukey's honestly significant difference post-hoc test, $P \leq 0.05$). (model implemented in R: lmer(variable ~ O₃ treatment $^*$ leaf position $^*$ time + mesophyll layer +(1 | cell/tree)).
(DOCX)

## Acknowledgments

We gratefully acknowledge technical and scientific support from Barbara Moura (image analysis), Terry Menard (microscopy), Christophe Robin (leaf physiology), Jean-Charles Olry and Stéphane Martin (PEPLor facility), and the Center for Microscopy and Image Analysis at the University of Zürich (electron microscopy).

## Author Contributions

**Conceptualization:** Benjamin Turc, Yves Jolivet, Mireille Cabané, Marcus Schaub, Pierre Vollenweider.

**Data curation:** Benjamin Turc.

**Formal analysis:** Benjamin Turc.

**Funding acquisition:** Yves Jolivet, Mireille Cabané, Marcus Schaub, Pierre Vollenweider.

**Investigation:** Benjamin Turc.

**Methodology:** Benjamin Turc, Pierre Vollenweider.

**Supervision:** Yves Jolivet, Mireille Cabané, Pierre Vollenweider.

**Visualization:** Benjamin Turc.

**Writing – original draft:** Benjamin Turc.

**Writing – review & editing:** Benjamin Turc, Yves Jolivet, Mireille Cabané, Marcus Schaub, Pierre Vollenweider.

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
