## [Decision Letter · Decision Letter 0]

22 Dec 2022

PONE-D-22-30841Ante- and post-mortem cellular injury dynamics in hybrid poplar foliage as a function of phytotoxic O3 dosePLOS ONE

Dear Dr. Benjamin Turc,

Thank you for submitting your manuscript to PLOS ONE. After careful consideration, we feel that it has merit but does not fully meet PLOS ONE’s publication criteria as it currently stands. Therefore, we invite you to submit a revised version of the manuscript that addresses the points raised during the review process.

We look forward to receiving your revised manuscript.

Kind regards,

Md Ashrafuzzaman, Ph.D.

Academic Editor

PLOS ONE

Journal Requirements:

"This work was supported by the French National Research Agency (ANR, “Investissement d’avenir” from the program Lab of Excellence ARBRE: ANR-11-LABX-0002-01, to YJ), and by a Swiss Federal Institute for Forest, Snow and Landscape Research (WSL) internal grant (201701N1428, to PV)."

Reviewers' comments:

Reviewer's Responses to Questions

**Comments to the Author**

1. Is the manuscript technically sound, and do the data support the conclusions?

Reviewer #1: Yes

2. Has the statistical analysis been performed appropriately and rigorously? 

Reviewer #1: Yes

3. Have the authors made all data underlying the findings in their manuscript fully available?

Reviewer #1: Yes

4. Is the manuscript presented in an intelligible fashion and written in standard English?

Reviewer #1: Yes

5. Review Comments to the Author

Reviewer #1: This a well-written article with a solid experimental design. The authors applied logical ozone dose, as in Asia, 80-100 ppb ozone is frequently measured in the field. It has an excellent description of the experiment with figures and data. I enjoyed reading this article and strongly recommend accepting it for publishing in PLOS ONE.

However, I have a minor suggestion regarding non-destructive reflectance-based chlorophyll estimation. See line 389. It might not be accurate as a biochemical assay, but many authors have already proved it as one of the best methods if three points are measured from a leaf. Also, repeated measurement allows one to understand plants' responses in different growth stages under stress without destroying the plants. Please see the bellow articles:

https://doi.org/10.1007/s11356-022-19282-z

https://doi.org/10.1016/j.envpol.2017.06.055

https://doi.org/10.1111/pce.13864

6. PLOS authors have the option to publish the peer review history of their article (what does this mean?). If published, this will include your full peer review and any attached files.

Reviewer #1: **Yes: **Muhammad Shahedul Alam

---

## [Author Response · Author response to Decision Letter 0]

5 Feb 2023

Regarding non-destructive reflectance-based chlorophyll estimation (line 389): I agree and added your suggestion to the manuscript. Reflectance-based estimation is a very good way to estimate chlorophyll content, and has this advantage to be non-destructive, allowing multiple measurements on the same leaf along stress. We used this method in the first experiment of this project (Turc et al, 2021; doi: 10.3389/fpls.2021.679852), recording 10 measurements per leaf. Although this second experiment was conducted in the exact same conditions as a previous one described in Turc et al (2021), chlorophyll content decreased earlier when using biochemical assay. Leaves of our poplar were very big, therefore biochemical estimation of chlorophyll content on homogenized half leaf might be more accurate than reflectance-based method (even if we did 10 records per leaf).

---

## [Decision Letter · Decision Letter 1]

7 Feb 2023

Ante- and post-mortem cellular injury dynamics in hybrid poplar foliage as a function of phytotoxic O3 dose

PONE-D-22-30841R1

Dear Benjamin Turc,

We’re pleased to inform you that your manuscript has been judged scientifically suitable for publication and will be formally accepted for publication once it meets all outstanding technical requirements.

Kind regards,

Md Ashrafuzzaman, Ph.D.

Academic Editor

PLOS ONE

Reviewers' comments:

Reviewer's Responses to Questions

**Comments to the Author**

1. If the authors have adequately addressed your comments raised in a previous round of review and you feel that this manuscript is now acceptable for publication, you may indicate that here to bypass the “Comments to the Author” section, enter your conflict of interest statement in the “Confidential to Editor” section, and submit your "Accept" recommendation.

Reviewer #1: All comments have been addressed

2. Is the manuscript technically sound, and do the data support the conclusions?

Reviewer #1: Yes

3. Has the statistical analysis been performed appropriately and rigorously? 

Reviewer #1: Yes

4. Have the authors made all data underlying the findings in their manuscript fully available?

Reviewer #1: Yes

5. Is the manuscript presented in an intelligible fashion and written in standard English?

Reviewer #1: Yes

6. Review Comments to the Author

Reviewer #1: The authors have addressed the comments, and I highly recommend publishing the manuscript. I enjoyed reading it once again.

7. PLOS authors have the option to publish the peer review history of their article (what does this mean?). If published, this will include your full peer review and any attached files.

Reviewer #1: **Yes: **Muhammad Shahedul Alam

---

## [Editor Report · Acceptance letter]

20 Feb 2023

PONE-D-22-30841R1 

*Ante*- and *post-mortem* cellular injury dynamics in hybrid poplar foliage as a function of phytotoxic O_3_ dose 

Dear Dr. Turc:

I'm pleased to inform you that your manuscript has been deemed suitable for publication in PLOS ONE. Congratulations! Your manuscript is now with our production department. 

Kind regards, 

on behalf of

Dr. Md Ashrafuzzaman 

Academic Editor

PLOS ONE